# A PRECONDITIONED ACCELERATED STOCHASTIC GRADIENT DESCENT ALGORITHM

## ABSTRACT

We propose a preconditioned accelerated stochastic gradient method suitable for large scale optimization. We derive sufficient convergence conditions for the minimization of convex functions using a generic class of diagonal preconditioners and provide a formal convergence proof based on a framework originally used for on-line learning. Inspired by recent popular adaptive per-feature algorithms, we propose a specific preconditioner based on the second moment of the gradient. The sufficient convergence conditions motivate a critical adaptation of the per-feature updates in order to ensure convergence. We show empirical results for the minimization of convex and non-convex cost functions, in the context of neural network training. The method compares favorably with respect to current, first order, stochastic optimization methods.

## 1 INTRODUCTION

Large scale optimization tasks require computationally fast algorithms and have strict memory requirements. This makes gradient descent methods a prime choice in many areas of science due to their simplicity and light computational burden. Motivated by the need to further lower the computational and memory load, stochastic gradient descent (SGD) methods (Robbins & Monro, 1951; Tieleman & Hinton, 2012; Kingma & Ba, 2014) have become the default choice of optimization algorithm for machine learning, deep learning and more generally large scale data processing (Deng et al., 2013; Schmidhuber, 2015). Various acceleration techniques have been devised to improve the convergence speed of SGD. They can be generally grouped into two classes. A first class of approaches use inertia or averaging techniques to improve the quality of the stochastic gradient (Sutskever et al., 2013; Nesterov, 1983). The second class applies preconditioning or per-feature adaptation to improve the overall conditioning of the optimization task, and thus the speed up the convergence (Tieleman & Hinton, 2012; Kingma & Ba, 2014; Duchi et al., 2011).

In this paper we propose a preconditioned accelerated stochastic gradient descent (PA-SGD) method with a generic bounded preconditioner and analyze its convergence properties for convex cost functions. The method combines both approaches by coupling Nesterov's accelerated gradient descent (Nesterov, 1983, NAGD) with a varying diagonal preconditioner and a stochastic approximation of the gradients. The convergence results assume that the algorithms step size and acceleration coefficient are decaying. For the preconditioner, a sufficient condition for convergence is a bound on the rate of variation between two consecutive steps.

The generic preconditioner can be understood as per-feature adaptation of learning rate similar to many popular methods such as ADAM (Kingma & Ba, 2014), RMSProp (Tieleman & Hinton, 2012) or NADAM (Dozat, 2016). In contrast, the update equations of popular per-feature adaptive algorithms, such as ADAM, do not directly control the rate of variation of the per-feature scaling and can have a divergent behavior (Sashank J. Reddi & Kumar, 2018). They add a small constant to the decaying exponential sum of the squared gradient to ensure numerical stability. Very recently, Sashank J. Reddi & Kumar (2018) proposed a long term memory for the per-feature adaptation to guarantee convergence for momentum methods. Note that our work is using a generic per-feature preconditioner coupled with Nesterov's accelerated gradient. It only shares similar requirements, namely a bounded variation for the preconditioner. NADAM also combines per-feature adaptation with Nesterov's accelerated gradient. It relies on the gradient momentum to provide an improved gradient for a Nesterov acceleration scheme and it does not precondition directly the instantaneous

gradient, as we propose here. NADAM uses exactly the same per-feature adaptation strategy as ADAM and thus inherits the same convergence problem. Huang et al. (2017) and Johnson et al. (2016) also provide experimental evidence that ADAM is outperformed by SGD due to poor convergence. Since in this work we reconcile ideas from ADAM and Nestrov, the added extra update rules for the per-feature adaptation can also help with ADAM's convergence on convex cases.

This paper is organized as follows. We continue by introducing the main details of the algorithm in Section 2. Thereafter, an analysis of the convergence for convex functions and the assumptions required for the algorithm's convergence are presented in Section 3. We provide extensive simulation results showcasing the capabilities of the new method for both convex and non-convex cost functions in Section 4. We compare the proposed PA-SGD algorithm with ADAM, AMSGrad (Sashank J. Reddi & Kumar, 2018), and the stochastic adaptation of NAGD, herein denoted as NASGD. Finally, in Section 5 we summarize our main contributions.

## 2 THE PA-SGD ALGORITHM

Our proposed method combines Nesterov's accelerated gradient descent algorithm (Nesterov, 1983) with a diagonal preconditioning matrix that is applied directly to the instantaneous gradient at each iteration $t$. To introduce the algorithm we assume a sequence of cost functions $f_t$ at each given instant $t$. Such functions can be introduced either through the sampled data, by using subset of the available data, or due to the inherent properties of a function $f$ which is sampled at each time instant $t$. We aim to find the solution $\boldsymbol{\theta}$ that minimizes $f$. Starting from Nesterov's accelerated gradient descent method (Sutskever et al., 2013; Nesterov, 1983) we derive a stochastic version

$$\begin{cases} \tilde{\boldsymbol{\theta}}_{t+1} &= \tilde{\boldsymbol{\theta}}_t + \boldsymbol{a}_{t+1} \\ \boldsymbol{a}_{t+1} &= \mu_t \boldsymbol{a}_t - \alpha_t \nabla f_t(\tilde{\boldsymbol{\theta}}_t + \mu_t \boldsymbol{a}_t), \end{cases} \tag{1}$$

where at each iteration we have access to the stochastic realization $f_t$. Here we denote by $\tilde{\boldsymbol{\theta}}_t$ the solution at iteration $t$. The vector $\boldsymbol{a}_t$ represents the accelerated gradient, that is used as a descent direction. We allow for a time varying step size $\alpha_t$ as well as for a time varying acceleration parameter $\mu_t$. By performing a change of variable $\boldsymbol{\theta}_t = \tilde{\boldsymbol{\theta}}_t + \mu_t \boldsymbol{a}_t$ in (1) and introducing the diagonal preconditioning matrix $\boldsymbol{P}_t$ with diagonal elements $p_{t,i}$, we arrive at the proposed preconditioned accelerated stochastic gradient descent method (PA-SGD)

$$\begin{cases} \boldsymbol{\theta}_{t+1} &= \boldsymbol{\theta}_t - \mu_t \boldsymbol{a}_t + (1 + \mu_{t+1}) \boldsymbol{a}_{t+1} \\ \boldsymbol{a}_{t+1} &= \mu_t \boldsymbol{a}_t - \alpha_t \boldsymbol{P}_t \boldsymbol{g}_t. \end{cases} \tag{2}$$

Here, for ease of notation, we denote $\boldsymbol{g}_t = \nabla f_t(\boldsymbol{\theta}_t)$, with each element denoted by $g_{t,i}$.

### 2.1 CONVERGENCE ASSUMPTIONS

To guarantee convergence on convex cases, we require the following sufficient assumptions: the diagonal elements of the preconditioning matrix $\boldsymbol{P}_t$ are positive and there exists a lower bound for them; the resulting preconditioned absolute values of the gradient are bounded. Formally we have

$$\begin{aligned} \frac{1}{p_{t,i}} &\leq C_p, \qquad p_{t,i} > 0, \quad \forall i \\ \|\boldsymbol{P}_t \, |\boldsymbol{g}_t| \, \|_\infty &\leq C_{pg}, \end{aligned} \tag{A1}$$

where we denoted by $|\boldsymbol{g}_t|$ the absolute value for each element in $\boldsymbol{g}_t$. Additionally, we require that the acceleration $\mu_t$ and step size $\alpha_t$ are decaying as

$$\begin{aligned} \mu_t &= \mu_0 \lambda^t, \qquad 0 < \lambda < 1 \\ \alpha_t &= \alpha_0 t^{-c}, \qquad 0 < c < 1, \end{aligned} \tag{A2}$$

where generally $\lambda$ is chosen close to 1 and $c$ is chosen close to $0.5$. These assumptions are used in other algorithms such as ADAM. In this work, we argue that for convergence on convex cases we need an extra assumption, namely we require the pre-conditioner to satisfy a bounded rate of change,

$$\alpha_t p_{t,i} \leq \alpha_{t-1} p_{t-1,i}, \qquad \forall i, \tag{A3}$$

as a sufficient condition for guaranteed convergence on convex cases. The requirement specified by (A3) represents a key difference between our pre-conditioner and the per-feature weights of ADAM (Kingma & Ba, 2014).

## 2.2 PRECONDITIONER DESIGN

There are several preconditioners that meet the assumptions (A1) and (A3). Motivated by the success of the per-feature adaptation, we propose the following preconditioner

$$
\begin{aligned}
p_{t,i} &= \left( \sqrt{\frac{v_{t,i}}{(1-\beta^t)}} + \varepsilon \right)^{-1} \\
\tilde{v}_{t,i} &= \beta v_{t-1,i} + (1-\beta) g_{t,i}^2 \\
v_{t,i} &= \max \left( \tilde{v}_{t,i}, \ s \frac{\alpha_t^2 (1-\beta^t)}{\alpha_{t-1}^2 (1-\beta^{t-1})} v_{t-1,i} \right),
\end{aligned}
\tag{3}
$$

as it contains two major well known techniques, namely the NASGD and adaptive per-feature methods such as ADAM. We note that for $s = 0$, the choice of preconditioning in ADAM is recovered (Kingma & Ba, 2014). However, the overall algorithmic updates remain different since here we use the preconditioner for a Nestrov's accelerated gradient scheme. The quantity $\varepsilon > 0$ represents a small constant that is needed for numerical stability to avoid $p_{t,i}$ increasing to $\infty$ if $v_{t,i}$ approaches 0. The exponential window averaging factor denoted by $\beta$ is configured such that $0 < \beta < 1$.

The stability parameter $s$ controls the rate of variation of the preconditioner. Intuitively, it is a balance between convergence rate and stability. We show in Section 3 that the preconditioner from (3), with $s = 1$, meets assumptions (A1) and (A3). For a convex cost function and $s < 1$, the updates do not necessarily converge depending on the choice of step size and acceleration parameter value. Along the same lines, Sashank J. Reddi & Kumar (2018) noted that ADAM and other algorithms that use similar per-feature adaptation such as NADAM are lacking a mechanism to control the rate of change, and as a result may diverge in convex cases. This contribution and Sashank J. Reddi & Kumar (2018) propose different strategies for the preconditioner and the overall algorithm.

## 3 CONVERGENCE PROPERTIES

Our proof concerns convex cost function. We analyze the convergence in the online learning framework proposed by Zinkevich (2003). A sequence of convex cost functions $f_t(\boldsymbol{\theta}_t)$ for $t = 1 : T$ is assumed. At each iteration $t$, we aim to estimate the parameters $\boldsymbol{\theta}_t$ with respect to $f_t$. The framework from Zinkevich (2003) considers the convergence of the sequence of estimates $\boldsymbol{\theta}_t$ and their associated cost functions $f_t$ with respect to the best solution $\boldsymbol{\theta}^* = \arg\min_\theta \sum_{t=1}^T f_t(\boldsymbol{\theta})$ by defining the regret function

$$
R(T) = \sum_{t=1}^T \left( f_t(\boldsymbol{\theta}_t) - f_t(\boldsymbol{\theta}^*) \right).
\tag{4}
$$

The regret function $R(T)$ maps the goodness of fit of the sequence $f_t(\boldsymbol{\theta}_t)$ compared to the ideal solution $\boldsymbol{\theta}^*$.

For convergence guarantees, we assume bounded gradients and bounded variation in the parameter estimates, similarly to Kingma & Ba (2014),

$$
\begin{aligned}
\|\boldsymbol{\theta}_t - \boldsymbol{\theta}^*\|_\infty &\le C_\theta \\
\|\boldsymbol{g}_t\|_\infty &\le C_g.
\end{aligned}
\tag{A4}
$$

It can be easily verified that the preconditioner defined by (3) satisfies the assumptions (A1) and (A3). Assumption (A4) combined with the use of $\varepsilon$ and the property that the quantity $v_{t,i} \ge (1 - \beta) g_{t,i}^2$ ensures (A1). A choice $s = 1$, as a sufficient condition for convergence in case of convex cost functions, and the use of the max operator ensure that (A3) is satisfied.

Proof outline: to prove convergence under assumptions (A1), (A2), (A3) and (A4), we show that $\lim_{T \to \infty} \frac{R(T)}{T} = 0$. To this end, we upper-bound $|a_{t+1,i}|$, and $g_{t,i}(\theta_{t,i} - \theta^*)$, and use these bounds to eventually find an upper bound for $R(T)$.

**Lemma 1.** *Given a convex function $f : \mathbb{R}^d \to \mathbb{R}$, then for any $\boldsymbol{x}, \boldsymbol{y} \in \mathbb{R}^d$ we have $f(\boldsymbol{y}) \ge f(\boldsymbol{x}) + \nabla f(\boldsymbol{x})^\dagger (\boldsymbol{y} - \boldsymbol{x})$.*

**Property 2.** *Under assumptions (A1), (A2), (A3), and for a given iteration $t$ and coefficient index $i$ of the accelerated gradient $|a_{t+1,i}|$ is bounded from above as*

$$
|a_{t+1,i}| \le \alpha_0 C_{pg} \left( t^{-c} + \frac{\lambda^t}{1 - \mu_0} \right).
\tag{5}
$$

*Proof.* From (2) we can expand $|a_{t+1,i}|$ such that

$$
\begin{aligned}
|a_{t+1,i}| &= & |-\alpha_t p_{t,i} g_{t,i} + \mu_t a_{t,i}| \\
&= & |-\alpha_t p_{t,i} g_{t,i} - \sum_{j=1}^{t-1} \alpha_j p_{j,i} g_{j,i} \prod_{k=j+1}^{t} \mu_k| \\
&\leq^{(A1)} & \alpha_t p_{t,i} |g_{t,i}| + \sum_{j=1}^{t-1} \alpha_j p_{j,i} |g_{j,i}| \prod_{k=j+1}^{t} \mu_k \\
&\leq^{(A1)} & \alpha_t C_{pg} + \sum_{j=1}^{t-1} \alpha_j C_{pg} \prod_{k=j+1}^{t} \mu_k \\
&\leq^{(A2)} & \alpha_0 C_{pg} \left( t^{-c} + \sum_{j=1}^{t-1} j^{-c} \prod_{k=j+1}^{t} \mu_0 \lambda^k \right) \\
&\leq & \alpha_0 C_{pg} \left( t^{-c} + \sum_{j=1}^{t-1} j^{-c} \mu_0^{t-j} \lambda^t \right) \\
&\leq & \alpha_0 C_{pg} \left( t^{-c} + \lambda^t \sum_{j=1}^{t-1} \mu_0^j \right).
\end{aligned}
\tag{6}
$$

We have used $\prod_{k=j+1}^{t} \lambda^k \leq \lambda^t$ since $0 < \lambda < 1$ and $j^{-c} \leq 1$. Replacing the geometric progression with its upper bound we arrive at (5). ∎

**Property 3.** *Under assumptions* (A1)*,* (A2)*,* (A4) *and for any given iteration $t$ and coefficient index $i$ the quantity $g_{t,i} (\theta_{t,i} - \theta^*)$ is bounded from above as*

$$
\begin{aligned}
g_{t,i} (\theta_{t,i} - \theta^*) &\leq & \frac{1}{2\alpha_0 p_{t,i}} \left[ t^c (\theta_{t,i} - \theta_i^*)^2 - t^c (\theta_{t+1,i} - \theta_i^*)^2 \right] + \\
&& \frac{C_p}{2\alpha_0} \left[ \alpha_0^2 C_{pg}^2 (1 + 2\mu_0)^2 \left( t^{-c} + 2\frac{\lambda^t}{1-\mu_0} + \frac{\lambda^{2t} t^c}{(1-\mu_0)^2} \right) + \right. \\
&& \left. 2\mu_0^2 \alpha_0 C_\theta C_{pg} \left( \lambda^{2t} + \frac{\lambda^{3t} t^c}{1-\mu_0} \right) \right].
\end{aligned}
\tag{7}
$$

*Proof.* From (2) we have

$$
\boldsymbol{\theta}_{t+1} = \boldsymbol{\theta}_t + \mu_t \mu_{t+1} \boldsymbol{a}_t - (1 + \mu_{t+1}) \alpha_t \boldsymbol{P}_t \boldsymbol{g}_t.
\tag{8}
$$

By explicitly writing (8) for a component $i$ and by subtracting from both sides of (8) the ideal solution $\boldsymbol{\theta}^*$ and squaring the resulting equality we get

$$
\begin{aligned}
(\theta_{t+1,i} - \theta_i^*)^2 =& (\theta_{t,i} - \theta_i^*)^2 + (-\mu_t a_{t,i} + (1 + \mu_{t+1}) a_{t+1,i})^2 + \\
& 2 (\mu_t \mu_{t+1} a_{t,i} - (1 + \mu_{t+1}) \alpha_t p_{t,i} g_{t,i}) (\theta_{t,i} - \theta_i^*).
\end{aligned}
\tag{9}
$$

Here we also used (2) to replace $\mu_t \mu_{t+1} \boldsymbol{a}_t - (1 + \mu_{t+1}) \alpha_t \boldsymbol{P}_t \boldsymbol{g}_t$ by $-\mu_t \boldsymbol{a}_t + (1 + \mu_{t+1}) \boldsymbol{a}_{t+1}$. Rewriting the equality to express $g_{t,i} (\theta_{t,i} - \theta^*)$ we arrive at

$$
\begin{aligned}
g_{t,i} (\theta_{t,i} - \theta^*) &= & \frac{1}{2(1+\mu_{t+1})\alpha_t p_{t,i}} \left[ (\theta_{t,i} - \theta_i^*)^2 - (\theta_{t+1,i} - \theta_i^*)^2 + \right. \\
&& \left. (-\mu_t a_{t,i} + (1 + \mu_{t+1,i}) a_{t+1,i})^2 + 2\mu_t \mu_{t+1} a_{t,i} (\theta_{t,i} - \theta_i^*) \right] \\
&\leq & \frac{1}{2(1+\mu_{t+1})\alpha_t p_{t,i}} \left[ (\theta_{t,i} - \theta_i^*)^2 - (\theta_{t+1,i} - \theta_i^*)^2 + \right. \\
&& \left. (\mu_t |a_{t,i}| + (1 + \mu_{t+1}) |a_{t+1,i}|)^2 + 2\mu_t \mu_{t+1} |a_{t,i}| |(\theta_{t,i} - \theta_i^*)| \right] \\
&\leq^{(A4)} & \frac{1}{2(1+\mu_{t+1})\alpha_t p_{t,i}} \left[ (\theta_{t,i} - \theta_i^*)^2 - (\theta_{t+1,i} - \theta_i^*)^2 + \right. \\
&& \left. (\mu_t |a_{t,i}| + (1 + \mu_{t+1}) |a_{t+1,i}|)^2 + 2\mu_t \mu_{t+1} C_\theta |a_{t,i}| \right].
\end{aligned}
\tag{10}
$$

Applying (5) and relying on (A2) produces

$$
\begin{aligned}
g_{t,i} (\theta_{t,i} - \theta^*) &\leq^{(5),(A2)} & \frac{1}{2(1+\mu_{t+1})\alpha_t p_{t,i}} \left[ (\theta_{t,i} - \theta_i^*)^2 - (\theta_{t+1,i} - \theta_i^*)^2 + \right. \\
&& \alpha_0^2 C_{pg}^2 \left( \mu_0 \lambda^t \left( t^{-c} + \frac{\lambda^t}{1-\mu_0} \right) + \right. \\
&& \left. (1 + \mu_0 \lambda^{t+1}) \left( (t+1)^{-c} + \frac{\lambda^{t+1}}{1-\mu_0} \right) \right)^2 + \\
&& \left. 2\mu_0^2 \alpha_0 C_\theta C_{pg} \lambda^t \lambda^{t+1} \left( t^{-c} + \frac{\lambda^t}{1-\mu_0} \right) \right] \\
&\leq^{(A1),(A2)} & \frac{t^c}{2(1+\mu_0\lambda^{t+1})\alpha_0 p_{t,i}} \left[ (\theta_{t,i} - \theta_i^*)^2 - (\theta_{t+1,i} - \theta_i^*)^2 + \right. \\
&& \alpha_0^2 C_{pg}^2 (1 + 2\mu_0 \lambda^t)^2 \left( t^{-c} + \frac{\lambda^t}{1-\mu_0} \right)^2 + \\
&& \left. 2\mu_0^2 \alpha_0 C_\theta C_{pg} \lambda^t \lambda^{t+1} \left( t^{-c} + \frac{\lambda^t}{1-\mu_0} \right) \right],
\end{aligned}
\tag{11}
$$

which, under assumptions (A1) and (A2), further reduces to (7). ∎

**Theorem 4.** *Under assumptions* (A1), (A2), (A3) *and* (A4) *the regret function $R(T)$ is bounded from above as*

$$
\begin{aligned}
R(T) &\leq \frac{C_p}{2\alpha_0} \sum_{i=1}^d \Big[ T^c C_\theta^2 + \alpha_0^2 C_{pg}^2 \left(1 + 2\mu_0\right)^2 \left( \frac{2}{(1-\mu_0)(1-\lambda)} + \frac{1}{(1-\mu_0)^2(1-\lambda^2)^2} \right) + \\
&\quad 2\mu_0^2 \alpha_0 C_\theta C_{pg} \left( \frac{1}{1-\lambda^2} + \frac{1}{(1-\mu_0)(1-\lambda^3)^2} \right) + \alpha_0^2 C_{pg}^2 \left(1 + 2\mu_0\right)^2 \sum_{t=1}^T t^{-c} \Big] \\
&\leq \mathcal{O}\left( T^c + \sum_{t=1}^T t^{-c} \right).
\end{aligned}
\tag{12}
$$

*Proof.* We construct an upper bound for (4) by relying on Lemma 1. As such we have

$$
f_t(\boldsymbol{\theta}_t) - f_t(\boldsymbol{\theta}^*) \leq \boldsymbol{g}_t^\dagger \left( \boldsymbol{\theta}_t - \boldsymbol{\theta}^* \right).
\tag{13}
$$

Writing (13) for each component $i$, summing for $t = 1 : T$, we have

$$
\begin{aligned}
R(T) &= \sum_{i=1}^d \sum_{t=1}^T g_{t,i} \left( \theta_{t,i} - \theta_i^* \right) \\
&\leq^{(7)} \sum_{i=1}^d \sum_{t=1}^T \frac{1}{2\alpha_0 p_{t,i}} \Big[ t^c \left( \theta_{t,i} - \theta_i^* \right)^2 - t^c \left( \theta_{t+1,i} - \theta_i^* \right)^2 \Big] + \\
&\quad \frac{C_p}{2\alpha_0} \Big[ \alpha_0^2 C_{pg}^2 \left(1 + 2\mu_0\right)^2 \left( t^{-c} + 2 \frac{\lambda^t}{1-\mu_0} + \frac{\lambda^{2t} t^c}{(1-\mu_0)^2} \right) + \\
&\quad 2\mu_0^2 \alpha_0 C_\theta C_{pg} \left( \lambda^{2t} + \frac{\lambda^{3t} t^c}{1-\mu_0} \right) \Big] \\
&\leq \frac{1}{2\alpha_0} \sum_{i=1}^d \Big[ \frac{1}{p_{1,i}} \left( \theta_{1,i} - \theta_i^* \right)^2 + \sum_{t=2}^T \left( \left( \frac{t^c}{p_{t,i}} - \frac{(t-1)^c}{p_{t-1,i}} \right) \left( \theta_{t,i} - \theta_i^* \right)^2 \right) - \\
&\quad \frac{T^c}{p_{T,i}} \left( \theta_{T+1,i} - \theta_i^* \right)^2 \Big] + \\
&\quad \frac{C_p}{2\alpha_0} \Big[ \alpha_0^2 C_{pg}^2 \left(1 + 2\mu_0\right)^2 \sum_{t=1}^T \left( t^{-c} + 2 \frac{\lambda^t}{1-\mu_0} + \frac{\lambda^{2t} t^c}{(1-\mu_0)^2} \right) + \\
&\quad 2\mu_0^2 \alpha_0 C_\theta C_{pg} \sum_{t=1}^T \left( \lambda^{2t} + \frac{\lambda^{3t} t^c}{1-\mu_0} \right) \Big].
\end{aligned}
\tag{14}
$$

Relying on (A4) and using (A3) written as $\frac{t^c}{p_{t,i}} - \frac{(t-1)^c}{p_{t-1,i}} > 0$ for $\alpha_t = \alpha_0 t^{-c}$, we have

$$
\begin{aligned}
R(T) &\leq^{(A2)} \frac{1}{2\alpha_0} \sum_{i=1}^d \Big[ C_\theta^2 + C_\theta^2 \sum_{t=2}^T \left( \frac{t^c}{p_{t,i}} - \frac{(t-1)^c}{p_{t-1,i}} \right) - \frac{T^c}{p_{T,i}} \left( \theta_{T+1,i} - \theta_i^* \right)^2 \Big] \\
&\quad + \frac{C_p}{2\alpha_0} \Big[ \alpha_0^2 C_{pg}^2 \left(1 + 2\mu_0\right)^2 \sum_{t=1}^T \left( t^{-c} + 2 \frac{\lambda^t}{1-\mu_0} + \frac{\lambda^{2t} t^c}{(1-\mu_0)^2} \right) + \\
&\quad 2\mu_0^2 \alpha_0 C_\theta C_{pg} \sum_{t=1}^T \left( \lambda^{2t} + \frac{\lambda^{3t} t^c}{1-\mu_0} \right) \Big] \\
&\leq^{(A3)} \frac{C_p}{2\alpha_0} \sum_{i=1}^d \Big[ T^c C_\theta^2 + \alpha_0^2 C_{pg}^2 \left(1 + 2\mu_0\right)^2 \sum_{t=1}^T \left( t^{-c} + 2 \frac{\lambda^t}{1-\mu_0} + \frac{\lambda^{2t} t^c}{(1-\mu_0)^2} \right) + \\
&\quad 2\mu_0^2 \alpha_0 C_\theta C_{pg} \sum_{t=1}^T \left( \lambda^{2t} + \frac{\lambda^{3t} t^c}{1-\mu_0} \right) \Big].
\end{aligned}
\tag{15}
$$

The resulting inequality can be bounded using the upper bounds for the geometric series $\sum_{t=1}^T \lambda^t \leq \frac{1}{1-\lambda}$ and $\sum_{t=1}^T \lambda^{2t} \leq \frac{1}{1-\lambda^2}$ and for the arithmetic-geometric series $\sum_{t=1}^T \lambda^{2t} t^c \leq \frac{1}{(1-\lambda^2)^2}$ and $\sum_{t=1}^T \lambda^{3t} t^c \leq \frac{1}{(1-\lambda^3)^2}$, respectively. This produces the bound from (12). ∎

**Corollary 1.** *Under the assumptions from Theorem 4 it follows that*

$$
\lim_{T \to \infty} \frac{R(T)}{T} = 0.
\tag{16}
$$

*Proof.* This result can be obtained directly from Theorem 4 using the fact that the divergence rate of the hyper-harmonic series $\sum_{t=1}^T t^{-c}$, for $0 < c < 1$, is proportional to $T^{1-c}$. Thus, if $0 < c < 1$ all terms from (12) grow with a rate slower than $T$. ∎

## 4 SIMULATIONS AND RESULTS

We study the convergence properties of the proposed PA-SGD method in comparison with ADAM (Kingma & Ba, 2014), AMSGrad (Sashank J. Reddi & Kumar, 2018) and NASGD (Sutskever et al., 2013; Nesterov, 1983) as defined in (1). For all simulations we use the MNIST hand written number database (Lecun et al., 1998). All experiments are performed in Matlab. Throughout the simulations,

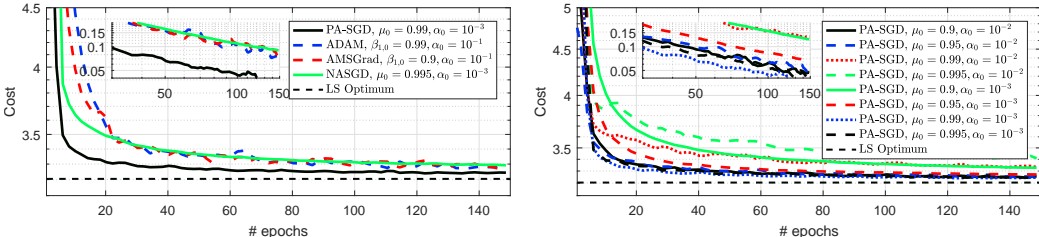

Figure 1: Convex cost function: evolution of the least squares cost for (left) the best performing parameter configuration of the tested algorithms; (right) several choices of acceleration parameter $\mu_0$ for PA-SGD. The new PA-SGD converges faster and achieves an almost optimal cost function value after $10 - 20$ epochs. ADAM with $\beta_{1,0} = 0.9$ and $\alpha_0 = 10^{-1}$ has similar convergence as AMSGrad and was omitted. We also include a zoom-in, in log-log scale, of the gap between the least squares cost and that of the other methods.

for all tested algorithms we use a stochastic mini batch of size $128$. We report the evolution of the cost function values across iterations. We present the best evolution of the tested algorithms, the configuration parameters being selected using a grid search. For better visibility we only present the convergence trend by averaging the results across multiple iterations.

All algorithms are configured to start from the same identical initialization, generated to be normally distributed with variance $0.1$. For ADAM and AMSGrad we use the same parameter names as presented by Kingma & Ba (2014) and Sashank J. Reddi & Kumar (2018), respectively, namely $\beta_{1,t}$ for the gradient momentum parameter, $\beta_2$ as the equivalent of $\beta$ from (3) in PA-SGD and $\alpha_t$ as the step size. The values for the gradient momentum are chosen $\beta = 0.999$ and $\beta_2 = 0.999$. The decay rate of the acceleration parameter from (A2), $\lambda = 1 - 10^{-8}$, and the numerical stability constant, $\varepsilon = 10^{-8}$, are kept constant across all experiments. The other individual configuration parameters are reported for each algorithm and for each test case.

We perform two classes of experiments. The first experiment set investigates the convergence on convex problems. In this setup we evaluate the performance of the algorithm under the assumptions that guarantee theoretical convergence. We perform a least squares regression directly with respect to the labels and a logistic regression experiment to classify digit $5$. The next experiments look into the empirical performance for the training of neural network models. In this case the cost functions are non-convex and the experiments fall outside the scope of our convergence proof. Here we use constant step sizes $\alpha$, a choice consistent with typical deep learning experiments. For our algorithm, the stability parameter $s$ is set to $0$. For the first practical experiment, we train a neural network with two $32$ neurons hidden layers to predict the MNIST numerical label. All activation functions are set to $\tanh$ and the labels are appropriately scaled. We use a mean squared error (MSE) as a cost function. The second experiment involves the training of similar neural network having two $32$ neuron hidden layers with ReLu (Hahnloser et al., 2000; Glorot et al., 2011) activation functions, resulting in a sub-gradient based optimization. We use this setup for a $\mathrm{softmax}$ multinomial logistic regression with respect to the $10$ classes in the dataset. Both problems exhibit sparse gradients which have been shown to be problematic for the stochastic gradient descent methods (Duchi et al., 2011).

### 4.1 CONVEX COST FUNCTIONS

For the experimental setup involving the convex cost functions we perform a grid search for the best parameters, for each algorithm, by varying the configuration parameters $\mu_0 \in \{0.9, 0.95, 0.99, 0.995\}$ for PA-SGD and NASGD, and $\beta_{1,0} \in \{0.9, 0.95, 0.99, 0.995\}$ for ADAM and AMSGrad and $\alpha_0 \in \{10^0, 10^{-1}, 10^{-2}, 10^{-3}, 10^{-4}\}$ for all algorithms. For the step size decrease, we use $c = 0.5$ in all simulations.

For the linear regression of the MNIST data, the cost function is convex and a closed form least squares (LS) solution exists. We use it as the optimal solution, to judge the behavior of the algorithms. In Figure 1, on the left, we present the results obtained with the best configuration for all algorithms. The convergence rate of the proposed method is better when compared to that of the

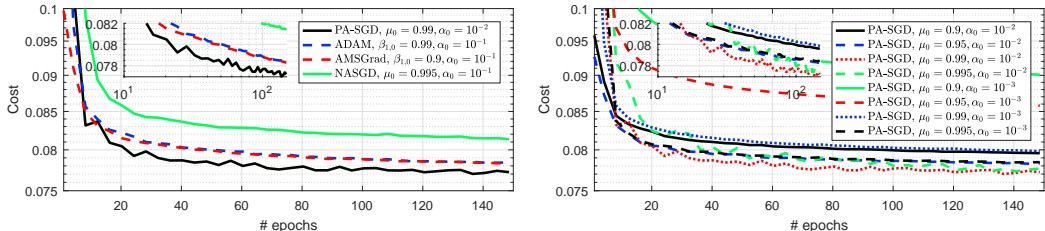

Figure 2: Convex cost function: evolution of the logistic regression cost for (left) the best performing parameter configuration of the tested algorithms; (right) several choices of acceleration parameter $\mu_0$ for PA-SGD. The new PA-SGD exhibits faster convergence than both ADAM and AMSGrad. ADAM with $\beta_{1,0} = 0.9$ and $\alpha_0 = 10^{-1}$ has similar convergence as AMSGrad and was omitted. We also include a zoom-in, in log-log scale.

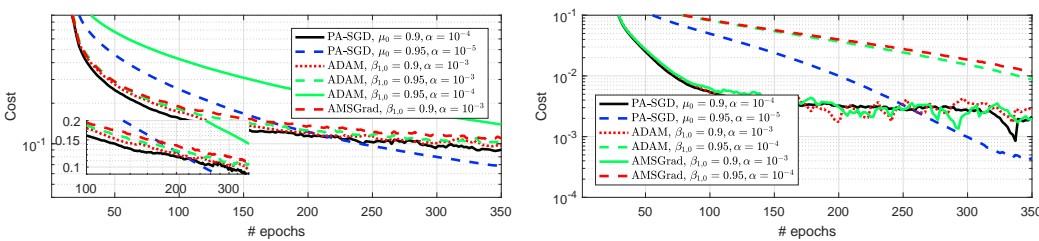

Figure 3: Neural network training: (left) evolution of the mean squared error cost; we also include a zoom-in, in log-log scale; (right) evolution of the multinomial logistic regression cost. The best performing parameter configurations are presented. For low value of the acceleration parameter $\mu_0$, PA-SGD exhibits comparable convergence with respect to ADAM and AMSGrad. Our method shows benefit when a larger acceleration parameter $\mu_0$ is used, which allows for a lower final cost albeit with slower initial convergence speed. AMSGrad has a comparable behavior with ADAM with all parameter choices. For the logistic regression we note that towards the final stages of convergence all methods exhibit large fluctuations of the cost function values.

other tested methods, reaching within $3\% - 4\%$ of the optimal LS solution in around 20 epochs. It shows the best performance for a larger acceleration parameters $\mu_0 = 0.99$ and $\alpha_0 = 10^{-3}$. For a better understanding of the convergence characteristics, we also present the convergence behavior as a function of the parameters $\mu_0$ in Figure 1, on the right, for two choices of the step size $\alpha_0$. The choice of $\mu_0$ moderates the convergence rate and its influence is coupled to that of the step size. If the step size is smaller, having a larger acceleration is beneficial, while for larger step sizes a very large acceleration is detrimental. The simulations involving the logistic regression task, presented in Figure 2, show a good behavior for our method, PA-SGD having a similar cost function value at 50 epochs as ADAM at 150. Our proposed method benefits from a smaller step size. We note that the convergence plots of ADAM and AMSGrad almost overlap.

In both experiments our method convergences at a faster rate than ADAM and AMSGrad. This is achieved for a smaller step size. Additionally, PA-SGD benefits from having a larger acceleration parameter $\mu_0$.

## 4.2 NEURAL NETWORK EXPERIMENTS, NON-CONVEX COST FUNCTIONS

For the experimental setup involving the neural network training we perform a similar grid search as before. We use a constant step size $\alpha$ using $c = 0$. For PA-SGD we set $s = 0$ in all simulations. Under this setup we only compare against ADAM and AMSGrad. We vary the acceleration parameter $\mu_0 \in \{0.9, 0.95, 0.99, 0.995\}$ for PA-SGD, momentum parameter $\beta_{1,0} \in \{0.9, 0.95, 0.99, 0.995\}$ for ADAM and AMSGrad and step size $\alpha \in \{10^{-2}, 10^{-3}, 10^{-4}, 10^{-5}, 10^{-6}\}$ for all algorithms.

In Figure 3 we compare the evolution for the proposed PA-SGD method for the two neural network training tasks. The first experiment involving the matching of the MNIST numerical labels in terms

of MSE our method compares favorably with respect to ADAM and AMSGrad. For the classification task, PA-SGD requires a lower step size and a larger acceleration to outperform the other methods in the later stages of the convergence. This has the drawback of a slower initial convergence. Again, as observed in both graphs, PA-SGD achieves a lower cost function value when a smaller step size is coupled with a larger acceleration parameter $\mu_0$.

A good practical range for the configuration parameters proved to be $\alpha \in [10^{-6}, 10^{-4}]$ and $\mu_0 \in [0.9, 0.99]$. In practice we have observed that a lower step size works best when coupled with an acceleration parameter $\mu_0$ closer to 0.99.

## 5 Conclusions

In this work, we proposed a preconditioned accelerated gradient method that combines Nesterov's accelerated gradient descent with a class of diagonal preconditioners, in a stochastic setting. We provided a regret bound, and we proved the algorithm's convergence for the minimization of convex cost functions. We also showcased empirically the properties of the algorithm for the minimization of convex and non-convex stochastic cost functions that occur usually in the context machine learning. The proposed PA-SGD method compares favorably with current stochastic optimization methods in terms of convergence speed while maintaining the same low computational complexity. This makes it well suitable for solving large scale, high dimensional optimization tasks.

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
