# OpenReview forum: "A preconditioned accelerated stochastic gradient descent algorithm"
_ICLR.cc/2019/Conference_

### Official Review · AnonReviewer1 · 2018-11-02
**see review**

**Rating:** 5
**Confidence:** 3

**Review:**

The paper talks about a method to combine preconditioning at the per feature level and Nesterov-like acceleration for SGD optimization.

The explanation of the method in Section 3 should be self-contained.  The main result, computational context, etc., are poorly described, so that it would not be easily understandable to a non-expert.

What was the reason for the choice of the mini batch size of 128.  I would guess that you would actually see interesting differences for the method by varying this parameter.

How does this compare with the FLAG method of Chen et al from AISTATS, which is motivated by similar issues and addresses similar concerns, obtaining stronger results as far as I can tell?

The figures and captions and inserts are extremely hard to read, so much so that I have to trust it when the authors tell me that their results are better.

The empirical evaluation for "convex problems" is for LS regression.  Hmm.  Is there not a better convex problem that can be used to illustrate the strength and weaknesses of the method.  If not, why don't you compare to a state-of-the-art least squares solver.

For the empirical results, what looks particularly interesting is some tradeoffs, e.g, a slower initial convergence, that are shown.  Given the limited scope of the empirical evaluations, it's difficult to tell whether there is much to argue for the method.  But those tradeoffs are seen in other contexts, e.g., with subsampled second order methods, and it would be good to understand those tradeoffs, since that might point to where and if a methods such as this is useful.

The conclusions in the conclusion are overly broad.

---

### Official Review · AnonReviewer2 · 2018-11-02
**Interesting idea**

**Rating:** 4
**Confidence:** 5

**Review:**

This paper presents a preconditioned variant of Nesterov's Accelerated Gradient (NAG) for use with Stochastic Gradients. The appears to be an interesting direction given that Nesterov's Acceleration empirically works better than the Heavy Ball (HB) method. There are a few issues that I'd like to understand:

[0] The authors make the assumption A 1-3, wherein,
- why should the momentum parameter mu drop exponentially? This is not required for the convex case, see Nesterov's (1983) scheme for the smooth case and the smooth+strongly convex case.
- why should the bounded rate of change (A3) even be satisfied in the first case? Does this hold true even in simple settings such as optimizing a quadratic/for log loss with stochastic and/or deterministic gradients? In short, is this a reasonable assumption (which at the least holds for certain special cases) or one that helps in obtaining a convergence statement (and which is not true even in simple situations)?


[1] Convergence under specific assumptions aside, I am not sure what is the significance of the regret bound provided. In particular, can the authors provide any reasoning as to why this regret bound is better compared to ADAM or its variants [Reddi et al, 2018]? Is there a faster rate of convergence that this proposed method obtains? I dont think a regret bound is reflective of any (realistic) practical behavior and doesn't serve as a means to provide a distinction between two algorithms. What matters is proving a statement that offers a rate of convergence. Other forms of theoretical bounds do not provide any forms of distinction between algorithms of the same class (adagrad, rmsprop, adam and variants) and are not reflective of practical performance.

[2] The scope of empirical results is rather limited. While I like the notion of having experiments for convex (with least squares/log loss) and non-convex losses, the experiments for the non-convex case are fairly limited in scope. In order to validate the effectiveness of this scheme, performing experiments on a suitable benchmark of some widely used and practically applicable convnet with residual connections/densenet for cifar-10/imagenet is required to indicate that this scheme indeed works well, and to show that it doesn't face issues with regards to generalization (see Wilson et al, 2017 - marginal value of adaptive gradient methods in machine learning).

[3] The paper claims of an "accelerated stochastic gradient" method - this really is a loosely used term for the paper title and its contents. There are efforts that have specifically dealt with acclerating SGD in a very precise sense which the paper doesn't refer to:
- Ghadimi and Lan (2012, 2013), Dieuleveut et al (2017) accelerate SGD with the bounded variance assumption.
- Accelerating SGD is subtle in a generic sense. See Jain et al. (2017) "Accelerating Stochastic Gradient Descent for Least Squares Regression", Kidambi et al. (2018) "On the insufficiency of existing momentum schemes for stochastic optimization". The former paper presents a rigorous understanding of accelerating SGD. The latter paper highlights the insufficiencies of existing schemes like HB or NAG for stochastic optimization.

---

### Official Review · AnonReviewer3 · 2018-11-06
**Combining Adam and Nesterov's momentum, but why?**

**Rating:** 4
**Confidence:** 3

**Review:**

Authors propose combining Adam-like per-feature adaptative and Nesterov's momentum. Even though Nesterov's momentum is implemented in major frameworks, it is rarely used, so there's an obvious question of practical relevance of proposed method.

Significant part of the paper is dedicated to proof of convergence, however I feel that convergence proofs are not interesting to ICLR audience unless the method is shown to be useful in practice, hence experimental section must be strong. Additionally, there's a veritable zoo of diagonal preconditioning methods out there already, this puts an onus on the authors to show an advantage in terms of elegance or practicality.

Experimental section is weak:
- There are 2 tunable parameters for each method. PA-SGD seems to be a bit better in the tail of optimization for convex problem, but I'm not confident that this is not due to better tuning of parameters (ie, due to researcher degrees of freedom). Authors state that "grid search was used", but no details on the grid search.
- PA-SGD seems quite sensitive to choice of alpha.
- No comparison against momentum which is probably the most popular method for neural network training nowadays (ie, ImageNet training is done using momentum and not Adam)
- non-linear optimization experiments are shown using logarithmic scale for y for a huge number of epochs. This amplifies the tail behavior. More relevant is measure like "number of steps to reach xyz accuracy", or wall-clock time
- it seems to perform equivalent to Adam for non-linear problem

---

### Meta-Review · Area_Chair1 · 2018-12-13
**AC decision**

**Confidence:** 4
**Recommendation:** Reject

**Metareview:**

Dear authors,

The reviewers pointed out a number of concerns about this work. It is thus not ready for publication. Should you decide to resubmit it to another venue, please address these concerns.